# What Is New on Paediatric Echocardiography for the Diagnosis, Management and Follow-Up of the Multisystem Inflammatory Syndrome Associated with COVID-19?

**DOI:** 10.3390/children9020146

**Published:** 2022-01-24

**Authors:** Matteo Di Nardo, Alessio Franceschini, Pierre Tissieres, Marcello Chinali

**Affiliations:** 1Pediatric Intensive Care Unit, Bambino Gesù Children’s Hospital, IRCCS, 00165 Rome, Italy; 2Department of Medical and Surgical Cardiology, Bambino Gesù Children’s Hospital, IRCCS, 00165 Rome, Italy; alessio.franceschini@opbg.net (A.F.); marcello.chinali@opbg.net (M.C.); 3Division of Paediatric Intensive Care and Neonatal Medicine, Paris South University Hospital, 78 Rue du General Leclerc, 94275 Le Kremlin-Bicetre, France; pierre.tissieres@aphp.fr

**Keywords:** pediatric lung ultrasound, SARS-CoV-2, pediatric critical care, COVID-19, MIS-C

## Abstract

Conventional echocardiography is an essential tool for the diagnosis, bedside management and follow-up evaluations of children with multisystem inflammatory syndrome associated with COVID-19. However, a more comprehensive echocardiographic exam, including myocardial deformation parameters, may allow early identification of subtle changes in ventricular function, provide risk stratification and, identify sub-clinical cardiac dysfunction at follow-up. Thus, myocardial deformation analysis should be routinely integrated to conventional echocardiography assessment in these patients.

## 1. Introduction

Between December 2019 and January 2020, a novel coronavirus (SARS-CoV-2) outbreak started in China and rapidly spread worldwide, leading the World Health Organization (WHO) to declare on 30 January 2020 a Health Emergency, and then, on 11 March 2020, a pandemic [1].

SARS-CoV-2 causes a highly contagious respiratory infectious disease (COVID-19), which, since the beginning, has been more frequent and severe in adults than in children [1,2]. Clinical manifestations in adults are variable and include fever and dyspnoea. However, the most severe cases can develop acute respiratory distress syndrome (ARDS) and multiorgan failure [2]. The clinical presentation of SARS-CoV-2 infection in children, instead, is generally mild and includes respiratory and gastrointestinal symptoms. COVID-19 progression towards more severe forms is possible, especially in children with comorbidities [3].

A stronger innate immune response, a lower number of angiotensin-converting enzyme (ACE) 2 receptors and the colonization of nasopharyngeal mucosa by other respiratory viruses may contribute to the lower severity of COVID-19 in children than in adults [4]. Unfortunately, this early enthusiasm has been reduced by the description of a new severe post-infection multisystem inflammatory syndrome among children (MIS-C or Paediatric Inflammatory Multisystem Syndrome temporally associated with SARS-CoV-2 infection, PIMS-TS) previously exposed to SARS-CoV-2 [5].

## 2. MIS-C

MIS-C generally develops a few weeks after SARS-CoV-2 infection and may present symptoms overlapping with Kawasaki disease (KD), toxic shock syndrome and macrophage activating syndrome. MIS-C incidence is currently unknown due to the fast evolution of the SARS-CoV-2 pandemic and its pathophysiology seems to be attributable to a hyperimmune response to the virus in a genetic susceptible child [5]. Common manifestations include persistent fever with elevated inflammatory markers and gastrointestinal symptoms; however, a minority of children may present skin rash, hands and/or feet erythema, conjunctivitis and cheilitis. More severe cases may present cardiac, respiratory and neurologic problems (Figure 1a,b) [6]. Treatment is empirical and aims to reverse the hyperinflammatory state with the use of intravenous immunoglobulins, steroids, and other immunomodulatory medications. Severe forms of MIS-C, presenting refractory vasodilatory/cardiogenic shock or severe arrhythmias, are managed in paediatric intensive care units (PICU) and may require extracorporeal membrane oxygenation [3].

## 3. Conventional Echocardiography for the Diagnosis, Management and Follow-Up of MIS-C

Conventional echocardiography is essential for a rapid evaluation of the cardiac function in children with MIS-C. Acute cardiac failure in patients with MIS-C is frequent (80–85% of cases) and is the major determinant of severity [6]. Cardiac manifestations include ventricular dysfunction, pericardial effusions, coronary artery dilation or aneurysms and arrhythmias. All these macroscopic findings are often associated with high levels of N-terminal prohormone of brain natriuretic peptide (NT-proBNP) and cardiac troponin [6,7].

Conventional echocardiography (Figure 2a–c), including left ventricle ejection fraction, mitral inflow velocities by spectral Doppler, tricuspid annular plane systolic excursion by M-Mode, and lateral mitral/tricuspid peak velocities on tissue Doppler, are useful to diagnose cardiac dysfunction, guide clinical decisions (fluid therapy, inotropic/mechanical circulatory support), monitor treatment response, and support follow-up programs in children with MIS-C [6,7]. Conventional echocardiography is also helpful to evaluate and follow coronary artery dilation and aneurysms. Both have reported a variable incidence (8–24%), according to the scores used to classify coronary dilation (Figure 2c) [6,7]. Currently, the clinical impact of coronary dilation is still poorly understood, and it could be considered as a transient response to increased myocardial oxygen consumption due to tachycardia and fever.

## 4. Myocardial Deformation Analysis

Recent data shows that a more comprehensive echocardiographic evaluation, including myocardial deformation parameters (Figure 2d–f), may allow the early identification of subtle changes in ventricular function, provide risk stratification for acute events and, identify subclinical cardiac dysfunction at follow-up in children with MIS-C [7]. Myocardial deformation analysis is obtained by integrating the data from the echocardiographic myocardial speckle signals, the tissue-blood border detection, the mitral annulus motion, and the periodicity of the cardiac cycle. It evaluates the deformation of the cardiac muscle during contraction. In systole, the myocardium reduces its wall length in both the longitudinal and circumferential planes, thus, more negative values of circumferential and longitudinal strains represent a better cardiac contraction. In diastole, the myocardium increases its wall length and the radial strain, thus a good left ventricular function is represented by a positive value of the radial strain (Figure 2g–i). Among the non-invasive technique for the diagnosis of myocarditis, cardiac magnetic resonance imaging is considered the gold standard, with a 79% accuracy rate compared with cardiac biopsy. Cardiac magnetic resonance imaging can detect myocardial oedema and/or fibrosis and is able to provide accurate strain analysis. However, accessibility to cardiac magnetic resonance imaging is often limited and cannot be performed in critically ill children. Nonetheless, recent evidence suggests that conventional echocardiography with regional myocardial deformation analysis can identify abnormalities, which are consistent with the oedematous regions found with cardiac magnetic resonance imaging, and can detect regional cardiac improvement or persistent cardiac function impairment due to residual fibrosis, found with cardiac magnetic resonance imaging [8].

MIS-C may present either as myocarditis with severe cardiogenic/distributive shock, or as myocarditis with preserved left ventricle ejection fraction, high BNP and cardiac troponin levels. In the latter case, deformation parameters [global longitudinal strain (GLS), left ventricle longitudinal systolic strain (LVLS), global circumferential strain (GCS), end-diastolic longitudinal strain rate (EDSR_L_), end-diastolic circumferential strain rate (EDSR_C_), left atrial strain (LAS) and right ventricle free wall longitudinal strain (RVFWLS)] are significantly below the normal ranges and should prompt therapeutic interventions (Figure 2g–i), which may include the early use of angiotensin-converting enzyme inhibitors [7,9,10,11].

Of note, in children with MIS-C and preserved left ventricle ejection fraction, a significant inverse correlation was found between NT-proBNP and troponin-I levels and GLS, GCS, LAS and RVFWLS [7], thus, in the absence of myocardial biopsies or of a cardiac magnetic resonance imaging, NT-proBNP and troponin-I may be suggestive for myocarditis [12]. Other groups [13] also reported that a severely impaired (≤−15.2%) LVLS at hospital admission was an independent predictor of PICU admission and extracorporeal membrane oxygenation support need [13]. Surprisingly, initial troponin levels and left ventricle ejection fraction at hospital admission, instead, were less sensitive [13].

## 5. MIS-C Follow-Up

Early follow-up studies with conventional echocardiography have shown that MIS-C patients returned to normal left ventricle ejection fraction within days or a couple of weeks [6]. However, a more comprehensive evaluation including deformation parameters has shown the persistence of subclinical systolic and diastolic dysfunction, even after 4–6 weeks from the acute phase [7,13]. Children with severely impaired (≤−13%) LVLS at hospital admission may present abnormal strain findings (LVLS, EDSR_L_) at 10 weeks follow-up, suggesting that a subclinical myocardial dysfunction is still present. Awareness of these findings may support prolonged pharmacological therapy with angiotensin-converting enzymes inhibitors, beta-blockers or mineral corticoid antagonists and guide continuous monitoring over time [13]. Follow-up data on coronary artery dilation and aneurysms are limited, but it appears that these coronary artery changes resolve over time [6].

## 6. Conclusions

A comprehensive clinical evaluation with echocardiography may be useful to support the clinical diagnosis of MIS-C, prompt the initiation of immunomodulatory interventions and cardiac treatments (e.g., angiotensin-converting enzyme inhibitors, mineralocorticoid antagonists and/or beta-blockers), and help to monitor their effect, informing follow-up care.

## Figures and Tables

**Figure 1 children-09-00146-f001:**
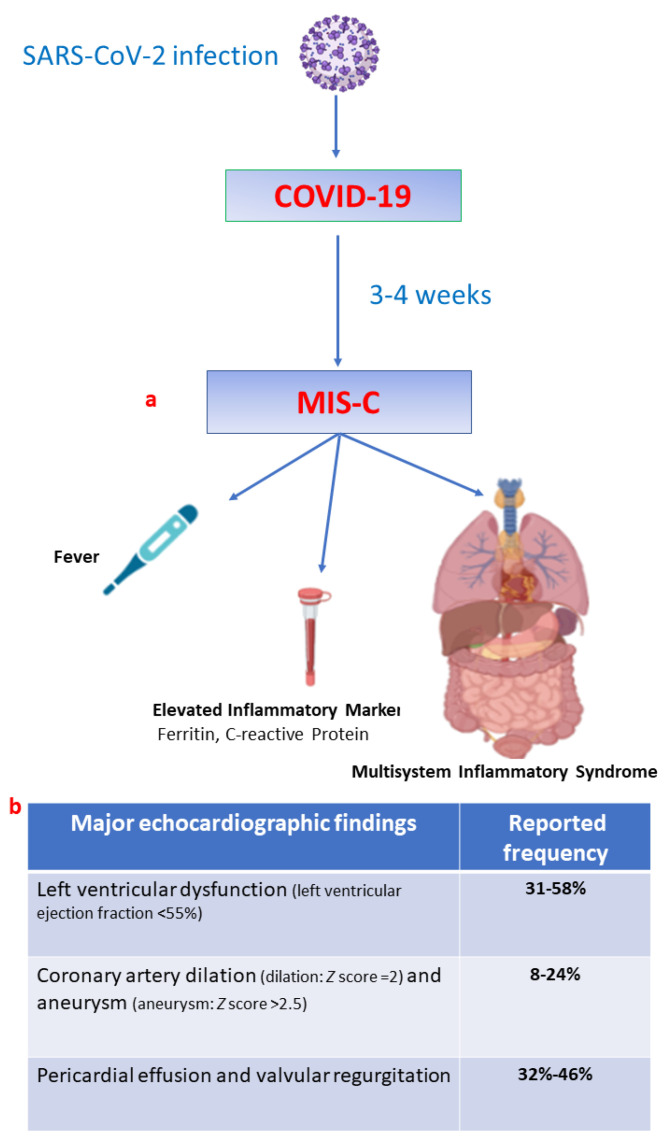
(**a**) Multisystem Inflammatory Syndrome associated with the main features of COVID-19 (**b**) prevalence of different echocardiographic findings in children with MIS-C (adapted from Alsaied T et al. [6], and Matsubara D. et al. [7]). MIS-C: multisystem inflammatory syndrome associated with COVID-19; AVC: aortic valve closure.

**Figure 2 children-09-00146-f002:**
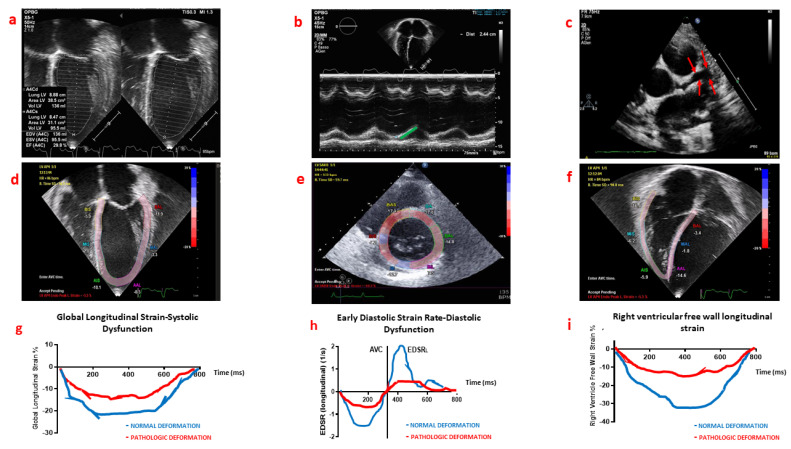
(**a**–**c**) Echocardiographic images of a patient with MIS-C with cardiac dysfunction (EPIQ 7, S8-3 paediatric cardiac probe, Philips Medical System, Milan, Italy), (**a**) left ventricular ejection fraction (Simpson’s biplane method), (**b**) tricuspid annular plane systolic excursion by M-mode (green line), (**c**) left anterior descending coronary artery aneurysm (Boston Z-score system: 3) (red arrow), (**d**–**f**) speckle tracking analysis images: left ventricular longitudinal and circumferential strain and right ventricular free wall longitudinal strain. (**g**) global longitudinal strain curves in a normal (blue line) and MIS-C children (red line) (**h**) early diastolic longitudinal strain rate curves (EDSR_L_) in a normal (blue line) and MIS-C children (red line) (**i**) right ventricular free wall longitudinal strain in a normal (blue line) and MIS-C children (red line). MIS-C: multisystem inflammatory syndrome associated with COVID-19; AVC: aortic valve closure.

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
