# Peer review of "What Is New on Paediatric Echocardiography for the Diagnosis, Management and Follow-Up of the Multisystem Inflammatory Syndrome Associated with COVID-19?"

_children, 2022, doi:10.3390/children9020146_

Round 1
Reviewer 1 Report
The authors present an overview of the latest echocardiographic techniques adopted for diagnosis and follow-up of patients with MIS-C. They report growing evidence supporting the need of speckle tracking echocardiography and deformation analysis of heart chambers in order to diagnose subclinical damage which could hel in identify patients at higher risk for worse outcome. Moreover such technique will be helpful during follow-up.
The message is important, as strain analysis should be included in the standard evaluation of MISC patients.
English language does not need significant review.
- I would consider to add a comment/little paragraph regarding cardiac MRI and the possible role for MRI strain analysis and LGE in this specific condition
Author Response
We are thankful with the Reviewer for reviewing our manuscript.
We fully agree with your suggestion and we added a part on cardiac resonance imaging to give a more comprehensive view of the problem. Please see the attachment

Reviewer 2 Report
Nice presentation of the impact and potential of cardiac ultrasound in COVID infected children!
A paragraph or statement presenting the differences to adult COVID patients would be beneficial.
Author Response
We are thankful with the Reviewer for Reviewing our manuscript.
We agree with the Reviewer and we included in our manuscript a paragraph discussing the main differences between adult and pediatric COVID-19. PLease see the attachment
